# Global Trends in Research on Biological Control Agents of *Drosophila suzukii*: A Systematic Review

**DOI:** 10.3390/insects16020133

**Published:** 2025-01-30

**Authors:** Lenon Morales Abeijon, Júlia Birkhan, Jana C. Lee, Sérgio Marcelo Ovruski, Flávio Roberto Mello Garcia

**Affiliations:** 1Programa de Pós-Graduação em Fitossanidade, Universidade Federal de Pelotas, Pelotas 96000, RS, Brazil; lenon.bio@gmail.com; 2Departamento de Ecologia, Zoologia e Genética, Instituto de Biologia, Universidade Federal de Pelotas, Pelotas 96000, RS, Brazil; juliabirkhan89@gmail.com; 3Crops Disease and Pest Management Research Unit, USDA-ARS, Corvallis, OR 97330, USA; jana.lee@usda.gov; 4Pilot Plant for Microbiological Industrial Processes and Biotechnology (PROIMI-CONICET), Biological Control Division, Tucumán T4001MVB, Argentina

**Keywords:** spotted-wing drosophila, biocontrol agents, entomopathogens, parasitoids, predators, natural enemies

## Abstract

This review reports the main developments in the biological control of spotted-wing drosophila (SWD), *Drosophila suzukii*, using parasitoids, predatory insects, and entomopathogens. We reviewed research conducted worldwide on the biological control of spotted-wing drosophila over the past decade (2012–2023). We examined 184 publications to see how many focused on each control agent, what methods they used, and where the research was carried out. Most research focused on tiny wasps that attack the pest, the most common being *Trichopria drosophilae* and *Pachycrepoideus vindemmiae*, with increasing interest in others like *Leptopilina japonica* and *Ganaspis kimorum*. Fifty-five publications examined entomopathogenic agents, while only 15 studied the predatory effects on SWD. Most publications were conducted in controlled environments like labs or greenhouses. The findings show that using these natural enemies can effectively reduce the pest population, especially conservation efforts that support these natural enemies in the environment. The current review is crucial for developing eco-friendly ways to protect crops and reduce reliance on chemical pesticides, benefiting farmers and the environment.

## 1. Introduction

The spotted-wing drosophila (SWD), *Drosophila suzukii* (Matsumura, 1931) (Diptera: Drosophilidae), has become a significant global pest in the last decade damaging berry crops [1] such as strawberry (*Fragaria* spp.) (Rosaceae), blackberry (*Rubus* spp.) (Rosaceae), blueberry (*Vaccinium* spp.) (Ericaceae), and raspberry (*Rubus* spp.) (Rosaceae) [2]. The species is mainly a pest of berries, such as cherry, raspberry, blueberry, mulberry, and strawberry [3] but also attacks grapes (*Vitis vinifera*) (Vitaceae), oranges (*Citrus* spp.) (Rutaceae) and guava (*Psidium guajava*) (Myrtaceae) [4,5,6].

The species is native to Southeast Asia and has been established in various regions worldwide, including North America [7], South America [8], North Africa [9], Central America [10], Europe [5], and Oceania [11]. Their widespread distribution is attributed to their high potential for dispersal [7], tolerance of a broad range of climatic conditions [5], and wide host range. The damage caused by SWD occurs due to the female’s ability to pierce the fruit’s epidermis to lay eggs in healthy fruits [12,13,14] and subsequently by the consumption of the mesocarp by the larvae, causing significant economic impact [5,15,16,17].

Economic loss was estimated by Bolda et al. [18] to have a potential impact of USD 511.3 million on strawberries, blueberries, raspberries, blackberries, and cherries in California, Oregon, and Washington, United States of America (USA). Also, in the USA, DiGiacomo et al. [19] reported losses of 20% of raspberry production in Minnesota. De Ros et al. [20] estimated losses of EUR 3.3 million in Trento, Italy. In South America, Morales [21] estimated production losses between USD 5000 and 17,550 per hectare for cherries and USD 4000 for blueberries in Ñuble, Chile. In Brazil, Benito et al. [22] estimated potential revenue losses of up to USD 21.4 million for peaches and USD 7.8 million for figs. In addition to production losses, pest control costs increased considerably, with a 37% drop in revenue in raspberries and 20% in strawberries in California due to increased insecticide costs [23].

Chemical control with pyrethroid, organophosphate, spinosyn, and neonicotinoid insecticides has been widely used to control SWD populations [24,25]. Spraying these pesticides adversely affects the environment and human health [11,26] and the possibility of generating resistance [27]. From this perspective, biological control of SWD is crucial to mitigate the economic and environmental damage caused by this invasive pest through sustainable management [28].

Therefore, this article aims to investigate the current status, limitations, and research opportunities on SWD’s biological control through a comprehensive review, highlighting successful cases. More specifically, we seek to answer the following questions: (1) What is the most studied biological control agent (BCA) group? (2) What are the most studied species contributing to SWD management? (3) What are the most common methods used in research (laboratory, greenhouse, or field)? (4) What is the commonly adopted scope of research (conservation, classical, augmentative, or survey)? (5) How was control success determined in biocontrol studies?

## 2. Materials and Methods

We conducted a comprehensive review on scientific articles following the guidelines of Preferred Reporting Items for Systematic Reviews and Meta-Analyses (PRISMA statement and checklist (Appendix A); http://www.prisma-statement.org) (accessed on 25 February 2024) [29,30]. All references of all publications found (see Figure 1; global flowchart) are available in Appendix A.

### 2.1. Search Strategy, Screening, and Eligibility Criteria

Two databases were used to search for published articles evaluating the use of BCAs on *D. suzukii*. The databases consulted were Scopus (from 1960 to 2023) and Web of Science Core Collection (1864 to 2023). Searches were performed in January 2024, using the terms: “spotted-wing drosophila” AND “biological control” OR “natural enem*” OR “parasitoid” OR “predator” OR “bacteria” OR “fungi” OR “virus” OR “nematode*”

The screening strategy included selecting publications in Portuguese, Spanish, and English and peer-reviewed journal materials. We did not use review articles, books, book chapters, conference papers, scientific notes editorials, and articles prior to 2012 and after 2023. Two criteria were used in the publication: (1) articles that include SWD; (2) articles that used one or more BCAs of SWD in a pest control context. No articles published before 2012 met the search criteria, justifying the initial period adjusted in the search.

Based on these eligibility criteria, the first search using the string terms resulted in 583 publication records, of which 399 publications were considered unfit to proceed with information extraction.

### 2.2. Data Extraction

Our search resulted in 184 publications suitable for review (see Appendix A). From the extracted data (digital object identifier “DOI”, title, abstract, and year of publication), it was possible to extract the following information: (1) the number of publications by BCA groups and species (parasitoids, predators, bacteria, fungi, viruses, nematodes, and multiple agents); (2) scope of the article, such as surveys including competition, interaction, laboratory trials, and natural/conservation, classical or augmentative biological control trials; (3) methodology used (laboratory tests, greenhouse conditions, field or combined); (4) evidence of the success of the BCA, obtained through the conclusion of the study on effectiveness, efficiency, and control; (5) countries where the article was developed. Competition, interaction, or laboratory evaluation studies were included as survey research. Diversity studies and new records of biological control agents were included as natural biological control articles. We used the location of the first author for each article without information about the research location. The nomenclature for the species of the *Ganaspis brasiliensis* complex was based on Sosa-Calvo et al. [31]. When the strain G1 was cited in a publication, the name *Ganaspis kimorum* was used; when G3 was cited, *Ganaspis lupini* was used, and when G5 was cited, *G. brasiliensis* was used. Species from China, Japan, and South Korea in which the strain was not mentioned were referred to as *G. kimorum* or *lupini*. In the other cases where the strain was absent, the geographic location was considered according to the distribution of these species.

### 2.3. Data Analysis

The graphs were generated using the “ggplot2” package [32] and with the addition of silhouettes from the “rplylopic” package [33] using R software version 4.3.2 [34].

## 3. Results

### 3.1. Publications Evaluating BCAs for SWD

Most publications on the biological control of SWD focused on parasitoids (64.1%), with the remaining groups comprising ≤ 10% each (Figure 2A). Among entomopathogens, publications focused on bacteria (10.3%), followed by fungi (6.5%), nematodes (4.9%) and viruses (4.4%) (Figure 2B). Publications on predators were less common (7.1%). In addition, five publications (2.7%) evaluated more than one BCA. Four publications combined nematodes with another BCA group (fungus, parasitoid, bacteria or predator). Only one publication assessed the use of parasitoids combined with predators.

#### 3.1.1. Parasitoids

Among the 120 publications on parasitoids (Hymenoptera) presented, 44 species were found to parasitize SWD. The Figitidae family was the most studied (46%; n = 21), followed by Braconidae (33%; n = 15) and Pteromalidae (13%; n = 6). The ten most studied SWD parasitoids worldwide were the pupal parasitoids *Trichopria drosophilae* (Perkins), *T. anastrephae* Lima (Hymenoptera, Diapriidae) and *Pachycrepoideus vindemmiae* (Rondani) (Hymenoptera, Pteromalidae) and larval *Ganaspis kimorum* Buffington, *G. brasiliensis* (Ilhering), *Leptopilina japonica* Novkovic and Kimura, *L. heterotoma* (Thomson) and *L. boulardi* Barbotin, Carton and Keiner-Pillault (Hymenoptera, Figitidae) and *Asobara japonica* Belokobylskij, *A. rufescens* (Förster*)* (Hymenoptera, Braconidae) (Figure 3).

#### 3.1.2. Entomopathogens

Twenty-three bacteria were recorded in publications on BCAs. Bacteria species such as *Xenorhabdus nematophila* (Poinar and Thomas), *Brevibacillus laterosporus* (Laubach), *Bacillus thuringiensis* Berliner (Bt), *Oenococcus oeni* (Garvie), *Leuconostoc pseudomesenteroides* Farrow, also presented promising publications. Eight families of viruses were reported infecting *D. suzukii*. La Jolla virus (Iflaviridae) appears in four publications. Other viruses, including Drosophila C virus (DCV) (Picornaviridae), Newfield virus (Permutotetraviridae), Drosophila A virus (DAV) and cricket paralysis virus (CrPV) (Dicistroviridae), Flock house virus (FHV) and Nagamuna virus (Nodaviridae), Sawaicho virus (Kitaviridae), Tama virus (Solemoviridae), Mogami virus (Chuvirus), and Notori virus (Phasmaviridae), were also investigated. Nine publications on nematodes evaluated as BCAs were also found. *Steinernema carpocapsae* (Weiser), *Heterorhabditis bacteriophora* Poinar, and *Steinernema feltiae* (Filipjev) were reported by the majority of articles. Among the entomopathogenic fungi, *Beauveria bassiana* Bals. (Vuill.), *Metarhizium anisopliae* (Metschn.) Sorokin and *Isaria fumosorosea* Wize stood out with seven, seven, and six publications, respectively.

#### 3.1.3. Predators

The review identified 15 species of predators from eight insect families. Of these, the families Anthochoridae (Hemiptera) and Carabidae (Coleoptera) stand out with four species each, and the species *Dolatia coriaria* (Kraatz) (Coleoptera: Staphylinidae) contained the most articles as a BCA. The minute pirate bug *Orius insidiosus* (Say, 1832) (Hemiptera: Anthocoridae) was studied in three publications, while green lacewing *Chrysoperla carnea* (Stephens) (Neuroptera: Chrysopidae) and European earwig *Forficula auricularia* L. (Dermaptera: Forficulidae) had two articles each. Other species, each with one publication, include the ground beetle *Bembidion quadrimaculatum* (L.), *Limodromus assimilis* (Paykull), *Poecilus cupreus* (L.), *Pterostichus melanarius* (Illiger) (Coleoptera: Carabidae), the true bugs *Dicyphus hesperus* Knight, *Macrolophus pygmaeus* (Rambur) and *Nesidiocoris tenuis* (Reuter) (Hemiptera: Miridae), *Orius laevigatus* (Fieber) and *Orius majusculus* (Reuter) (Hemiptera: Thripidae), *Podisus maculiventris* (Say) (Hemiptera: Pentatomidae), and *Gryllus pennsylvanicus* Burmeister (Orthoptera: Grillidae).

### 3.2. Evaluation of Publications on Biological Control Agents of Spotted-Wing Drosophila

#### 3.2.1. Methodology and Scope of Studies

Regarding methodologies, the majority of publications had experiments conducted in the laboratory (66%), while 15% of the articles were field studies. In addition, 18% of the publications adopted a combined approach, integrating multiple research contexts for a broader and more applicable understanding. Only 1% of the publications were in greenhouses (Figure 4).

More than half of the publications (58%) conducted surveys for SWD natural enemies (Figure 4). The remaining biological control approaches (42%) were categorized as natural/conservation biological control (26%; n = 48), which aimed to understand better and conserve natural ecological processes. In addition, 11% of the publications (n = 20) focused on classical biological control strategies, while 5% (n = 9) explored the potential of augmentative biological control.

Over the years, the number of SWD biological control research publications has steadily increased. Only two publications were recorded in 2012, 2013, and 2014. However, from 2015 and 2016 onwards, a significant increase was observed, with the number of articles increasing to eight each year. This growth continued consistently in the following years, with a notable increase in 2021, reaching 31 publications (Figure 5).

#### 3.2.2. Evidence of Success

Regarding the efficacy of SWD biological control, approximately 60.9% (n = 112) of the publications showed that the BCA mitigated damage caused by the pest, thus encouraging the development of more efficient control strategies. However, 39.1% (n = 72) of the publications did not demonstrate efficacy, either due to the intricate nature of the research or the lack of effective results.

Many publications focused on identifying potential BCAs, and 63.6% (n = 117) of the publications did not report on the efficiency of the BCAs. However, the remaining 36.4% (n = 67) examined efficiency. Among these, 6% showed results of potential control < 30% against the pest, 7% showed potential control between 30 and 50%, and 23% were identified as successful, achieving > 50% control of SWD.

Laboratory studies revealed the highest average efficiency (59.1%), with a relatively moderate variation reflected in a standard deviation of 23.47. In contrast, field studies revealed an average efficiency of 42.94% with a slightly higher variation (standard deviation of 26.89) (Table 1). Both efficiency values (minimum and maximum) ranged from 7% to 93%. Greenhouse studies revealed an intermediate average but with greater observed variability, represented by a standard deviation of 39.84 due to the low sample size.

Furthermore, the methodological analysis of the publications revealed that research in controlled environments demonstrated greater efficiency in evaluating the potential of SWD biocontrol agents, as evidenced by the significantly higher number of publications (Figure 6A). These publications make up the majority of the global literature on the subject. They may demonstrate the difficulty of transposing the results from the controlled environment to the natural environment of the pest. Furthermore, regarding the scope of these studies (Figure 6B), research that sought to identify biocontrol agents and those that used specimens from the environment itself (natural/conservation biological control) yielded more publications with successful efficiency as well as greater efficiency for parasitoids (Figure 6C).

Considering the total number of publications that recommended the use of the BCA (termed effective in Table 2), *T. drosophilae* (14.6% of publications) and *P. vindemmiae* (11.5%) parasitoids currently stand out as the main biological control agents.

The other BCAs were involved, at most, in 10% of studies that provided effective control. In addition, *T. drosophilae* and *P. vindemmiae* had more than a third of the publications rating them with >50% efficiency, and also showed higher percentages of parasitism at 38.6% and 34.2%, respectively. They were more effective and efficient in research in controlled environments (laboratory) than under natural conditions (field). Additionally, other control agents, such as *B. bassiana* and *I. fumusorosea* (93% efficiency) and the nematodes *H. bacteriophora*, *S. feltiae*, and *S. carpocapsae* (90% efficiency), were also efficient under these controlled conditions. In field conditions, greenhouse, or combined approaches, *T. drosophilae* showed the best results among the publications.

#### 3.2.3. Countries with Research on BCAs Against SWD

Research on the biological control of SWD was concentrated in 20 countries (Figure 7), with 41% having been conducted in Europe, 40% in North America, 11% in Asia, and 9% in South America. Of the total research found in the databases, the USA produced a quarter of all publications (25.5%; n = 47), followed by Italy (9.2%; n = 17), Germany (8.7%; n = 16), and Switzerland (7.6%; n = 14). The main countries of the studies that research SWD in South America are Argentina (4.9%) and Brazil (3.8%). In Asia, China (6.5%) and Japan (3.3%) have recorded research on SWD in that continent. Central America, Africa, and Oceania did not record any publications, as the invasion of SWD is very recent in these continents.

## 4. Discussion

Biological control is a safe and sustainable technology for pest management that uses natural enemies present in agroecosystems [35]. Here, we demonstrate this by quantifying publications that use BCAs to manage SWD throughout the world.

### 4.1. Frequently Reported Parasitoids for SWD Biological Control

More than half of the publications were about using parasitoids as biological control agents for SWD. Most studies to-date included solitary drosophilid pupal parasitoids in the families Pteromalidae and Diapriidae [28]. Pteromalidae are ectoparasitoids that oviposit in the hemocoel, between the puparium and the pupa [36], and *Muscidifurax raptorellus* Kogan and Legner, *P. vindemmiae*, *Spalangia erythromera* Forster, *Spalangia simplex* Perkins and *Vrestovia fidenas* (Walker) are known to parasitize SWD pupae. The Diapriidae are endoparasitoids, and *Phaenopria* spp., *T. anastrephae*, and *T. drosophilae* parasitized SWD [28].

*Trichopriae drosophilae* and *G. kimorum* have shown potential in controlling SWD populations. *Trichopriae drosophilae* can locate and parasitize the fly in several crops, including blueberries, cherries, and raspberries, both in the laboratory and the field [37,38]. Similarly, *G. kimorum* has reduced SWD populations in semi-field and field experiments [39]. In this regard, the greater specificity of *G. kimorum* against *D. suzukii* [40,41] has led several governments, such as Italy, Mexico and the USA, to approve releases of the parasitoid strain G1 [38,42,43,44,45,46]. Initial release efforts in Italy have confirmed the parasitoid’s ability to disperse, hibernate, and parasitize in the field [46]. Given recent releases of *G. kimorum* in Europe and the USA and the prevalence of the adventive larval–pupal parasitoid *L. japonica* in North America [47], these two species will likely become the focus of future publications.

In North America and Europe, research on native pupal ectoparasitoids has shown that *P. vindemmiae* and *T. drosophilae* parasitized, respectively, between 53% and 60% and between 38% and 76% of SWD pupae under laboratory conditions [48,49]. However, there are few publications focused on other native North American parasitoids that can utilize SWD pupae as hosts [50], and most of the parasitoids tested against SWD are not yet commercially available. Thus, it is essential to identify other effective native parasitoids that are easily accessible to berry growers in North America [51].

In addition, publications on parasitism in SWD highlight the potential of native and adapted parasitoids to control the pest. For example, the experimental adaptation of native parasitoids, such as specific pupal parasitoids, showed that the parasitism rate significantly increased after only three generations. In addition, the intrinsic competition between different parasitoids, such as *P. vindemmiae* and *T. anastrephae*, is explored to understand their dynamics [52,53]. Regarding parasitism capacity, Rossi-Stacconi and collaborators [37] showed a significant reduction (93%) in SWD emergence in field tests following the release of *T. drosophilae*.

In this review, eight parasitoid species were commonly studied, with at least ten published studies: *A. japonica*, *G. kimorum*, *L. heterotoma*, *L. japonica*, *T. anastrephae*, *T. drosophiliae*, and *P. vindemmiae*. These parasitoids differ by biological, physiological, and ecological factors and geographical distribution. Therefore, the applicability of each method varies according to the situations in which the pest is established. Among these physiological factors, it is interesting to investigate the different dietary sources of *Drosophila* spp. Parasitoids occur in various habitats, with the provision of multiple sources of sugar, such as tree sap, honeydew, and flowers, but mainly the food of the hosts, that is, fruit [54,55,56] The benefits associated with different sources of feeding for *T. drosophilae* and *P. vindemmiae* parasitoids were studied by Collatz and Romeis [57]. They offered flowers, infested and non-infested fruit, SWD hosts and honey droplets, or only honey; they showed that all the food sources extended the life expectancy of these parasitoids, justifying that one of the possible reasons for this effect is based on the provision of energy so that the female can parasitize the host.

Despite efforts to use *T. drosophilae* in North America and Europe for their ability to recognize SWD-infested and non-infested fruit in field and semi-field conditions [58], recent research in South America has focused on the pupal parasitoid *T. anastrephae*. They explored several aspects of the biocontrol agent, including the ability to parasitize under different laboratory conditions [52,59,60], competition with other parasitoids [53] and the toxicological effects of insecticides and essential oils [61]. Female *T. anastrephae* can recognize infested fruit or in stages of excessive ripeness, with larvae and pupae of the pest in strawberry crops. This parasitoid is capable of parasitizing the pest in infested strawberries and has a preference for fruit infested with eggs, larvae, and pupae compared to healthy and undamaged fruit [62].

Thus, the benefits of using parasitoids include their ability to attack different forms of SWD in the natural environment. Future efforts should focus on mass-producing these parasitoids, optimizing release strategies, as well as describing the potential risk of introduction in each country [11].

### 4.2. Frequently Reported Entomopathogens

Among the bacteria, *X. nematophila* and *B. thuringiensis* showed promising results in laboratory tests [28]. Hiebert et al. [63] searched for the composition and impact of bacteria associated with SWD, intending to develop effective and sustainable biological control strategies. They found the most effective bacteria to be *L. pseudomesenteroides*, known for its ability to block food intake in the SWD larvae, *Brevibacterium frigoritolerans* Delaporte and Sasson, as pathogenic for larvae and leading to a high mortality rate, and *Bacillus simplex* Priest et al., 1989 and *Bacillus altitudinis* Shivaji et al., 2006 both with deleterious effects on fly survival after oral infection. The identification of these microbes through different infection routes [64] can offer viable paths for candidates for biological control through survival analysis due to the pathogenic load provided by the control agent.

The main entomopathogenic fungi used in SWD control are *B. bassiana*, *M. anisopliae*, and *I. fumosorosea*. These fungi are evaluated by their ability to infect and kill SWD under controlled conditions [65]. The authors did not find virulence in SWD inoculated with *M. anisopliae* but did with *B. bassiana* under laboratory conditions. However, the application of *B. bassiana* in the field was not promising for virulence against SWD. Despite this, other studies indicated no significant differences in pathogenicity and virulence levels between *M. anisopliae* and *B. bassiana* [66]. When evaluated from the perspective of exposure time and adhesion to the SWD cuticle, the fungi may have a beneficial effect. Galland et al. [67], assessing five entomopathogenic fungi exposed to the pest for 10 s, 1 min, 10 min, 1 h, or 3 h, found that *B. bassiana* was more efficient, killing 50% of flies in four days with a 3 h exposure, while it took six days to achieve the same result with 10 s of exposure and ten days for the fungus to be 95% lethal to SWD individuals.

The nematodes tested against SWD that demonstrated potential, especially in laboratory settings, were *S. carpocapsae*, *H. bacteriophora*, and *S. feltiae*. Furthermore, *S. carpocapsae* has some efficacy in controlling SWD by infecting and killing larvae inside infested fruits [68]. However, its effectiveness can be influenced by soil type, humidity, temperature, and other environmental factors. In this sense, the SWD fly infection by the nematode *Steirnenema rarum* (De Doucet) PAM 25 to 14 °C at a concentration of 4.000 IJs/mL seems to significantly reduce the longevity of the pest [69]. In addition, efficacy may also be associated with how the pest’s immune pathways respond to the presence or absence of a nematode–bacteria complex. In some cases, *S. carpocapsae* is not recognized by the immune system of SWD larvae without the release of bacteria and the activation of genes that lead to the expression of antimicrobial peptides, rendering the cellular response inactive during infection [70].

Although fewer studies have focused on viruses, some research has identified several viral families that can infect SWD. The application of viruses as control agents is still in its early stages and requires further investigation to determine their practical use in the field. Recent research has identified La Jolla virus (LJV) (Picornavirales: Iflaviridae) as a possible biocontrol tool, showing pathogenic effects after oral administration, with infection causing reduced adult survival and interrupting larval development during pupation. In addition, LJV remained stable and infectious under different pH and temperature conditions, demonstrating its potential as an effective biological control agent against SWD [71]. Viruses, such as Newfield (NFV), can cause significant declines in female fertility through infection of follicular epithelium [72].

### 4.3. Frequently Studied Predators

Predators of SWD belong to nine families in the orders Coleoptera, Dermaptera, Hemiptera, Neuroptera, and Orthoptera. Predators such as minute pirate bugs in the family Anthocoridae and beetles in the family Carabidae have been investigated for their potential to control SWD [73]. Moreover, *Orius* spp. (Anthocoridae) and certain carabids can effectively prey on SWD under laboratory and field conditions [74]. Predators such as earwigs, spiders, and ants contribute significantly to the reduction in SWD populations, particularly in semi-natural habitats such as hedgerows, where they play a crucial role in the population dynamics of the pest [75]. The predator *Dolatia coriaria* (Coleoptera: Staphylinidae) has shown promise due to its ability to feed on several stages of the pest, with a maximum consumption capacity of 26 first-instar larvae, 15 second-instar larvae and six third-instar larvae [76]. The authors emphasize promoting natural predators in agroecosystems from the perspective of sustainable integrated pest management, thus aiming to reduce inputs within rural properties.

### 4.4. Research Trends and Future Perspectives

Most studies on the BCAs of SWD have focused on the identification and characterization of natural enemies of this pest. Laboratory studies dominate the research literature, representing about 66% of all studies, while field studies and combined approaches make up the remainder. Tests carried out in controlled environments help describe the potential of biocontrol agents before conducting greenhouse or field research [5,28]. Laboratory studies can first confirm the efficacy potential of SWD’s natural enemies, which are then evaluated for impact in the field, i.e., the earwig (*Forficula auricularia*) was assessed in the lab and field [77].

Furthermore, studies on the specificity and efficacy of parasitoids using combined laboratory and field tests can provide important information relevant to classical biological control [28]. Fellin et al. [46] suggested expanding studies on the release of the Asian G1 lineage of *Ganaspis* cf. *brasiliensis*, currently recognized as *Ganaspis kimorum* Buffington, 2024 [31], since pre- and post-release evaluations revealed that the parasitoid was recaptured in 50% of the release sites in Trento, Italy. The parasitoid *G. kimorum* emerged mainly from fresh fruits still on the plant, survived the climatic conditions imposed by winter, and revealed no adverse effects on non-target species.

One of the main challenges in the research of BCAs for SWD is translating laboratory results into field applications. While controlled experiments are crucial for evaluating the potential efficacy of BCAs, environmental factors such as temperature, humidity, and ecological interactions often confound efficacy in natural conditions. Furthermore, the lack of comprehensive studies in diverse cultivation environments, such as greenhouses and complex agricultural systems, limits the practical applicability of these findings.

Another significant challenge is the competition between biological agents, which can reduce their effectiveness when multiple BCAs are released simultaneously, necessitating more detailed studies on compatibility and the management of intraguild interactions. Wang et al. [35] showed that *L. japonica* generally outcompetes other parasitoids when multiple parasites attack a single host, depending on host conditions and density, emphasizing the need to consider the possibility of these interactions in management programs.

Other parasitoids also compete interspecifically, such as *P. vindemmiae* attacking pupae previously parasitized by *T. anastrephae* in the first instar stage [78]. Some competitors may have a physiological advantage, either by toxic secretion, induction of anoxia or nutritional deprivation or due to physical mechanisms [79,80,81], such as the sickle-shaped mandibles of *T. anastrephae* over the immaturity of *P. vindemmiae* [53].

Our review also identified a tendency of studies to focus on efforts to maintain natural enemies from the perspective of conservation biological control. Predators were demonstrated to remove a substantial proportion of SWD larvae in fruits or pupae in soil via predator inclusion/exclusion experiments [74,82]. Among efforts to encourage BCAs, an “augmentorium” [83] has been designed to rear parasitoids in-field by confining the pest while allowing the escape of parasitoids. This showed promise in field tests specific for SWD control, sustaining important native parasitoids and possibly associating with *T. drosophilae* releases [37,84].

In addition to technical and biological aspects, it is crucial to consider the environmental and social impacts of biocontrol strategies for SWD. BCAs can reduce the need for chemical pesticides, promote more sustainable agriculture, and minimize environmental damage. Studies have shown that the adoption of biological control can also benefit the health of agricultural workers and nearby communities by decreasing exposure to toxic substances [85,86]. Therefore, evaluating environmental and social impacts should be integral to biocontrol research to ensure that recommended practices are effective, sustainable, and safe.

Research on BCAs for SWD is distributed globally, with significant contributions from Europe (41.8%) and North America (39.7%), followed by Asia (9.8%) and South America (8.7%). The United States alone represents a substantial portion of this research. Studies from Asia and South America are also contributing to the growing body of knowledge on this subject [28].

Our review also identified that an important aspect is the diversity of biological control agents studied in different regions. For example, in Europe, much of the research has focused on parasitoids such as *G. kimorum* and *T. drosophilae*, while in North America, in addition to parasitoids, there is a significant focus on natural predators and nematodes. This diverse focus reflects the regional approaches and needs for controlling SWD [87]. It is essential to discuss how biological control agents adapt to each region’s local conditions. Studies in Asia have shown how local parasitoids are effective in reducing SWD populations under specific climatic conditions [29,40,88], while in South America, research focuses on the use of agents that can withstand high temperatures and humidity [8,89].

Another relevant point is the importance of international collaborations. Collaborative projects between European and American institutions have generated significant advances in developing biological control strategies. These partnerships facilitate the exchange of knowledge, technologies, and resources, thereby increasing the effectiveness of research and implementation control programs [87,90]. Finally, the implementation of research networks integrating scientists, agricultural producers, and policymakers can accelerate the development and adoption of more effective and sustainable biological control strategies for SWD management.

## 5. Conclusions

The review of biological control agents for *Drosophila suzukii* revealed insights into future advances, limitations, and perspectives. Entomopathogens such as bacteria, fungi, nematodes, and viruses have potential but are influenced by environmental factors and the pest’s immune response. Parasitoids, especially *Trichopria drosophilae* and *Ganaspis kimorum*, were effective under laboratory and field conditions, standing out as strong candidates for control programs.

Laboratory studies dominate, accounting for 66% of all research. This limits the applicability of findings to real-world conditions. Field studies are necessary to evaluate BCA performance under environmental variability and ecological complexity. Bridging this gap is crucial for the successful implementation of biocontrol strategies.

Regional research emphasized the importance of local adaptations. In Europe, *G. kimorum* and *T. drosophilae* were the primary focus. North American studies included predators and nematodes alongside parasitoids. Asian studies demonstrated the effectiveness of local parasitoids in specific climates, while South America focused on *T. anastrephae* for its resilience to heat and humidity. The most studied entomopathogens were the fungus *Beauveria bassiana*, *Metarhizium anisopliae*, and *Isaria fumosorosea*. Only *Beauveria* showed high efficiency under controlled conditions. Viruses demonstrated potential for biological control by reducing adult survival and interrupting the larval development of *Drosophila suzukii*.

Future research should prioritize field experiments to validate BCAs under natural conditions. Studies on the compatibility and synergy of combining parasitoids with predators or pathogens are essential. Research on intraguild predation must also advance. Strengthening international collaboration is key to sharing knowledge and developing effective global strategies against *D. suzukii*.

## Figures and Tables

**Figure 1 insects-16-00133-f001:**
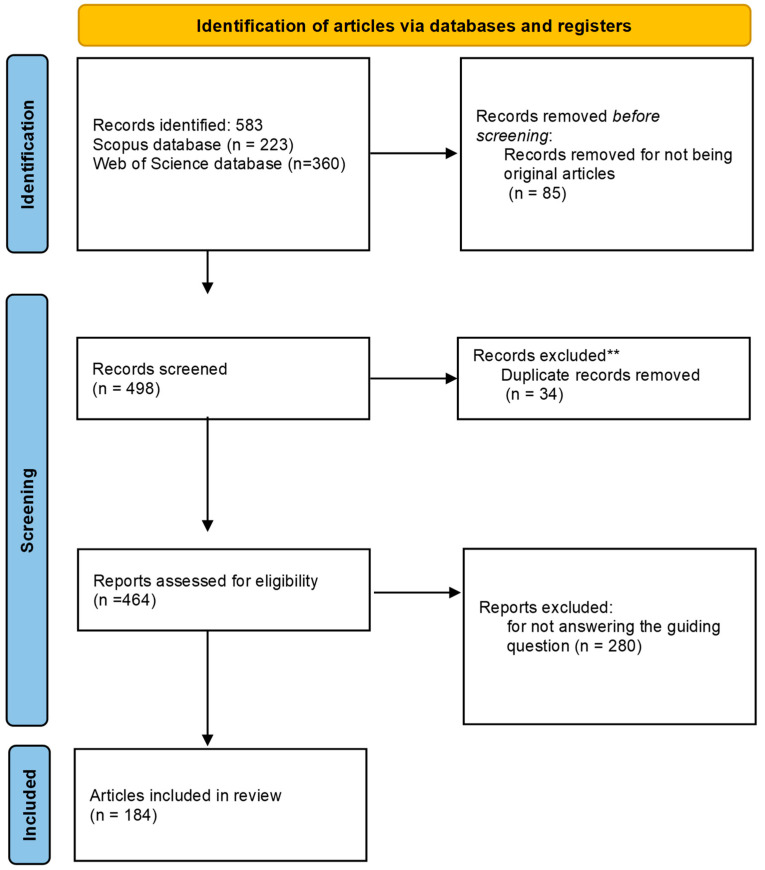
Flowchart of the PRISMA 2020 recommendation for selecting articles for systematic review on the use of biological control agents for *Drosophila suzukii*. ** Records exclude for being present in both databases.

**Figure 2 insects-16-00133-f002:**
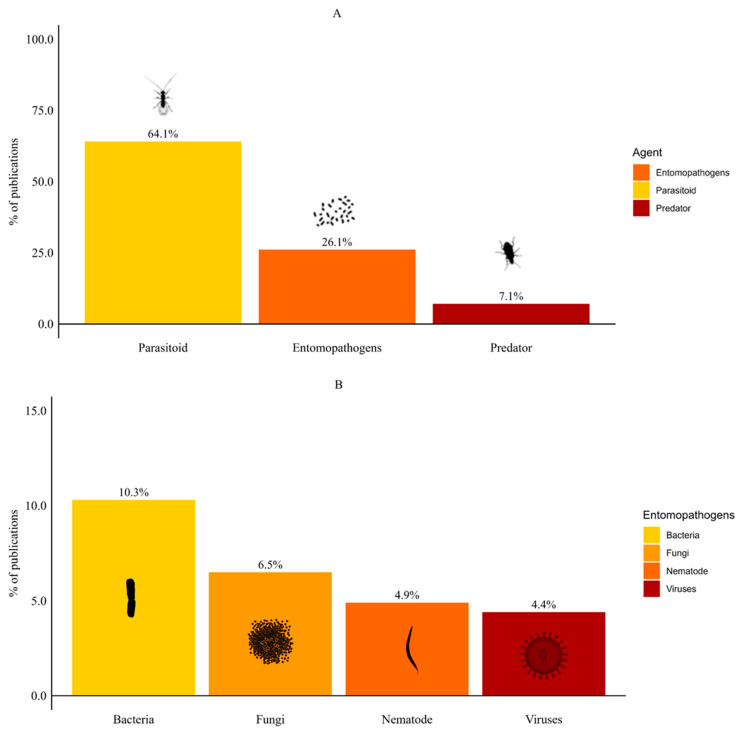
Types of biological control agents for spotted-wing drosophila. (**A**) Comparison among parasitoids, entomopathogens, and predators. (**B**) Distribution of entomopathogens into bacteria, fungi, nematodes, and viruses.

**Figure 3 insects-16-00133-f003:**
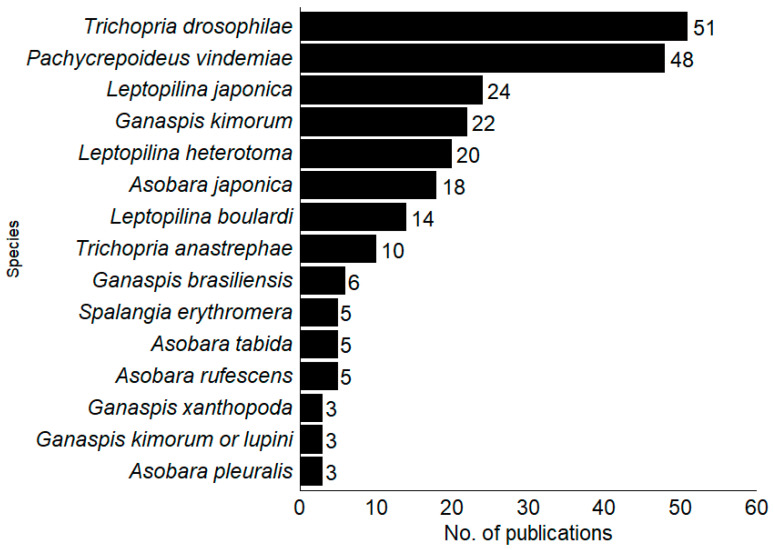
Parasitoid species have the highest number of articles, with the percentage of studies focused on each parasitoid species worldwide.

**Figure 4 insects-16-00133-f004:**
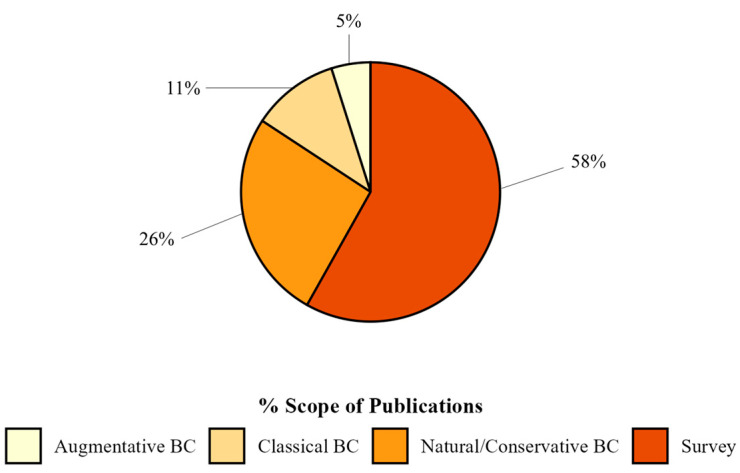
Percentage of publications grouped by scope on biological control of spotted-wing drosophila found in Web of Science and Scopus databases.

**Figure 5 insects-16-00133-f005:**
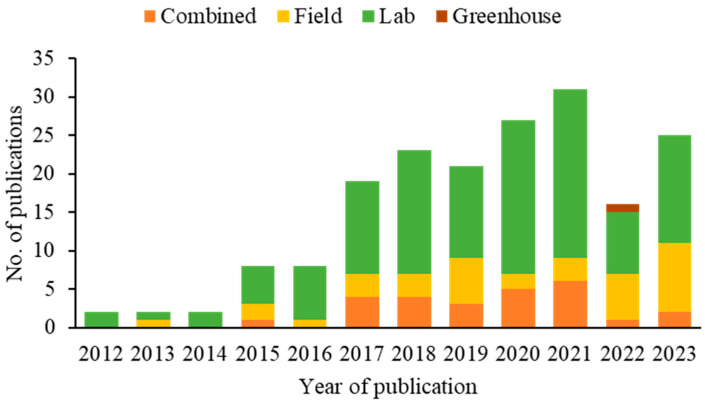
Methodological approaches applied in publications on biocontrol agents of spotted-wing drosophila (2012–2023).

**Figure 6 insects-16-00133-f006:**
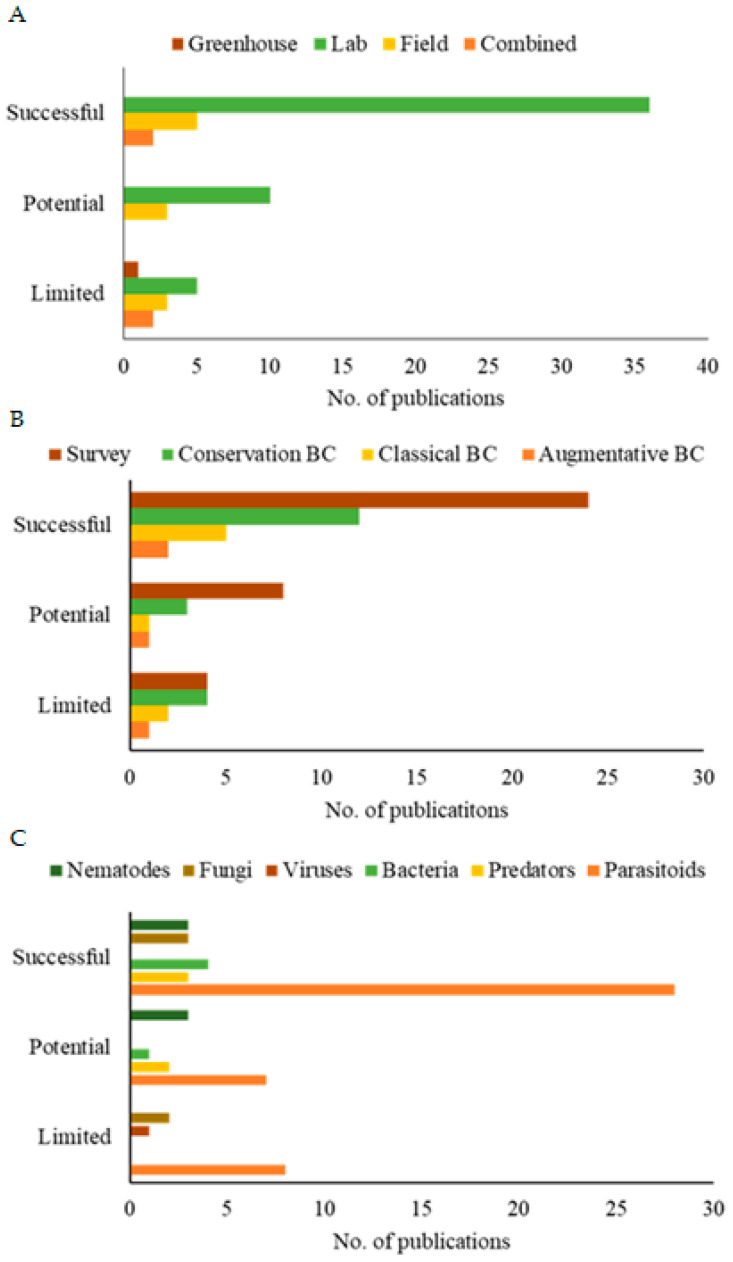
Publications on the efficacy of controlling spotted-wing drosophila using different methodological approaches (**A**), scope of study (**B**), and types of biological control agents (**C**) from 2012 to 2023. Efficiency parameters: limited (<30%); potential (30–50%); successful (>50%).

**Figure 7 insects-16-00133-f007:**
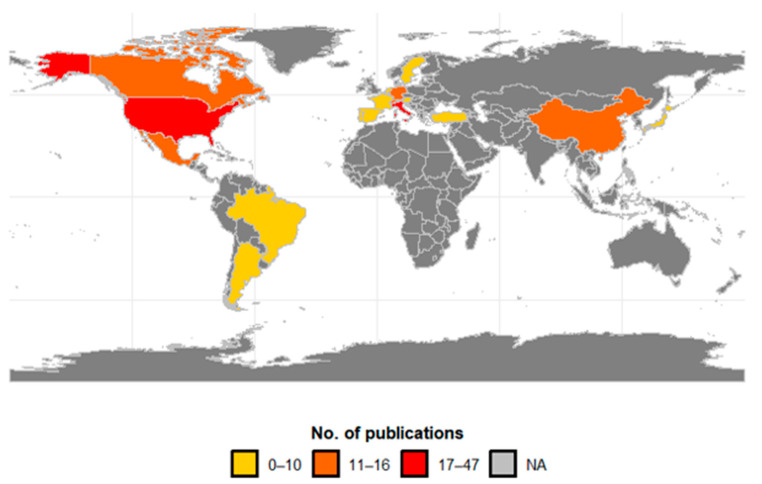
Geographical distribution of publications on biocontrol agents for spotted-wing drosophila by country. Publication intervals are represented by the colors: 0–10 (light yellow), 11–16 (orange), 17–47 (red), and NA (gray) indicating the absence of records.

**Table 1 insects-16-00133-t001:** Efficiency in different approaches found in spotted-wing drosophila publications extracted from the Web of Science and Scopus databases.

Approach	Number of Publications	Mean (%)	Standard Deviation	Variance	Min. Efficiency (%)	Max. Efficiency (%)	Median (%)
Lab	59	59.01	23.47	550.78	7.00	98.09	64.00
Field	13	42.94	26.89	722.86	7.00	93.00	34.09
Greenhouse	3	48.06	39.84	1587.09	17.09	93.00	34.08
Combined	8	45.89	31.50	992.64	7.00	93.00	47.04

**Table 2 insects-16-00133-t002:** Percentage of publications demonstrating effectiveness (E1), efficiency (E2) (>50%), and effectiveness and efficiency in different approaches of the 10 most effective *Drosophila suzukii* biological control agents found in studies extracted from the Web of Science and Scopus databases.

BC Agents	E1 (%)	E2 >50%	Approach Effectiveness (%)				Approach Efficiency (%)			
			Lab	Field	Greenhouse	Combined	Lab	Field	Greenhouse	Combined
*Asobara japonica*	3.6	14.0	7.1	0.0	0.0	0.0	12.7	0.0	0.0	0.0
*Asobara tabida*	1.2	2.6	0.9	0.0	0.0	1.8	0.0	0.0	0.0	3.6
*Ganaspis brasiliensis*	0.8	5.3	0.9	0.9	0.0	0.0	1.8	0.0	0.0	0.0
*Ganaspis kimorum*	3.2	15.8	5.3	0.9	0.9	0.0	3.6	1.8	0.0	1.8
*Leptopilina heterotoma*	4.7	15.8	7.1	2.7	0.0	1.8	14.5	0.0	0.0	3.6
*Leptopilina japonica*	2.8	18.4	3.5	2.7	0.0	0.0	7.3	1.8	0.0	3.6
*Leptopilina boulardi*	3.6	12.3	3.5	2.7	0.0	1.8	5.5	0.0	0.0	3.6
*Pachycrepoideus vindemmiae*	11.5	34.2	15.0	7.1	0.0	4.4	18.2	0.0	0.0	3.6
*Trichopria anastrephae*	2.0	7.9	3.5	0.9	0.0	0.0	5.5	0.0	0.0	0.0
*Trichopria drosophilae*	14.6	38.6	21.2	5.3	1.8	7.1	20.0	0.0	0.0	5.5

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
