# Peer review of "Global Trends in Research on Biological Control Agents of Drosophila suzukii: A Systematic Review"

_insects, 2025, doi:10.3390/insects16020133_

Round 1

Reviewer 1 Report

Comments and Suggestions for Authors

December 05, 2024

Dear Authors,

I send you my comments about the MS Current status, success cases, limitations and opportunities of Drosophila suzukii biological control: a systematic review (ID: insects-3355079). This is an excellent paper where the authors report the advances that have been made in biological control applied against D. suzukii, a pest that is spreading throughout various areas of the world and where it is attacking several species of berries. This systematic review considered the biological control with parasitoids, predators and entomopathogenic microorganisms, highlighting the most relevant aspects of each group of biological control agents.

The paper is very well written with excellent discussion. I have no major comments, I just made a few that the authors should address or clarify before being published in Insects.

Comments for the authors:

1.- Pag. 2.- The text between paragraphs 55 and 66 is not clear.

Please reword it in a way that is easier to express. Could you please prepare a table considering the loss by country or continent, crop species and the estimated amount lost?

2.- P5; Fig. 2A.- The bars in the figure indicating parasitoids and predators are the same colour. To differentiate, please change the colour of one of the bars.

3.- P7; L212-216.- They must refer to a table or figure where what is described in this text can be observed.

4. P7; L217-222.- Authors should refer to a table or figure where what they are describing in this text can be seen more clearly.

5.- P11; L318.- Publications intervals are represented by the colours, in the interval form 30- 40 is blue, but it is not indicated on the map.

6. P12; L399.- Priest et al. and Shivaji et al. are authors, the year of publication is missing, and they are also missing citation in the references section.

7.- P13; L407.- M. anisopliae, in italics.

P15, L531-542.- In conclusion, the authors should also mention the role that entomopathogens could play as biological control agent (BCA) of spotted-wing drosophila (SWD). Give arguments if they have potential to suppress the pest.

General comment:

1.- To cite the references consulted, authors must do so in accordance with the editorial standards of the Journal.

I really enjoyed reviewing this manuscript and I am available for any questions or comments you may have.

Comments on the Quality of English Language

The quality of English is good but can be improved.

Author Response

Comments 1: The authors have incorporated the majority of the changes. However, the references and the formatting of the headings are still not as per the MDPI format.

The revised title is incorrect: Replace it with: Global Trends in Research on Biological Control Agents of Drosophila suzukii: A Systematic Review

or: Global Research Trends on Biological Control Agents of Drosophila suzukii: A Systematic Review

Response 1: We agree with this comment. We incorporated the reference formatting. We included the suggested title, maintaining: "Global trends in research on biological control agents of Drosophila suzukii: a systematic review."

Comments 2: L276: Table heading: It is not BCA Agents; write BCAs or simply BC Agents.

Response 2: Agree. We modified the table heading to "BC Agents."

Reviewer 2 Report

Comments and Suggestions for Authors

The authors have made an excellent effort in writing this systematic review. This review is especially needed to recognize the potential biological control agents of spotted wing drosophila at the global scale. My comments are:

I would suggest revising the title: Current status, success stories, challenges, and opportunities in the biological control of Drosophila suzukii: a systematic review

L14: Be specific. Start with this systematic review reports the biological control. It does not explore biological control. The original studies have already been explored. You are only documenting or reporting the experiments done in the past.

L15: Instead of tiny wasps, mention parasitoids.

L16: Replace: We reviewed global research conducted on the biological control of spotted wing drosophila over the past decade (2012–2023).

L17-18: Be specific. Write the exact number of publications.

L20-21: Be specific. Write the number of publications.

L24: Add: The current systematic review…

L27-28: Starting lines of your abstract should focus on the background. Write clearly and also create a case of why you are conducting this systematic review.

L29: No need to capitalize all letters of Scopus.

L30: Our goal was to. Add “to”.

L32: Replace: Effectiveness of BCAs.

General comment: Intext citation style and references should be formatted as per the MDPI Insects style as mentioned in the instructions for authors.

L44: Only berry crops?

L49-50: Rewrite for more clarity.

L50-51: Mainly a pest of berries but also attacks…

L51-55: It could be written in a more straightforward and precise way. It has a lot of redundant phrases. Rewrite for clarity.

L57-66: U$ is incorrect. Use this format: US$5,000.

L76: Replace “state of the art” with “current status”.

L78-82: Replace: More specifically, we seek to answer the following questions: (1) What is the most studied biological control agent (BCA) group? (2) What are the most studied biological control species contributing to SWD management? (3) What are the most common methods used in research (laboratory, greenhouse, or field)? (4) What is the commonly adopted scope of research (conservation, importation, augmentation, or survey)? (5) How was control success determined in biocontrol studies?

Note: You can also use: conservation, classical and augmentative biological control, or survey?

L101-103: Figure 1: Inside the boxes, realign the text so that it looks decent. If you choose the left alignment, the follow that carefully.

L101: No need to capitalize the first letter of recommendations.

L141: Delete: “based on two criteria”. As you have already mentioned in the start.

 L141-145: Rewrite for clarity.

Figure 2: Try to write the description under x axis in a straight line. You can reduce font size.

Figure 2: Use different colors for the bars of predators and parasitoids. Use some good colors. Also, in figure 2B, try to create different colors.

Figure 2 caption: Don’t use the abbreviations in the figure captions. See figure 1 caption for more details.

L169-170: You can abbreviate the genus name Leptopilina after its first use.

Figure 3 caption: Don’t use abbreviations. A figure caption should be complete and independent enough to communicate a clear and complete message to the reader.

L178: bacteria species.

L180: Where does this sentence end?

L189: reported by majority of articles.

L207-208: Revise for clarity.

Figure 4: Revise figure caption for abbreviated content.

L233-235: Revise for clarity.

L240-242: Revise for clarity.

L253: Table title needs revision: same issue of abbreviated content.

Table 1: Column heading: Maximum efficiency. If you want to go with Max. then also write Min. Be consistent.

General comment: Figure and table captions should be as per the MDPI Insects format. I have not seen the use of – in the template.

Table 2: Revise table title for more clarity.

Table 2: Write complete scientific names. Don’t abbreviate.

L277-278: Use an asterisk, or a couple of asterisks, or simply 1 and 2. This is not actually labelled as per the MDPI Insect guidelines.

L295-296: Revise this sentence.

Figure 5,6: Don’t use abbreviations.

Figure 5B: Conservation, Importation (or you can use Classical Biological Control) and augmentation. Survey is fine.

Figure 5C: You are using plural form for Bacteria, Viruses, Fungi but singular for Nematode, Predator and Parasitoid. Revise it for consistency.

Figure 5 caption and complete paper: You can use limited (<30), Moderate (30-50%) and successful (>50%). Instead of limited, you can also use minimal or inadequate.

General comment: Double check the format of heading, sub-heading and sub-sub-heading.

L302-309: It is irrelevant. You need to start with a summary of your findings, rather than reporting a conference. You can add this somewhere in the beginning when you are building the case for this systematic review. It is not a good fit here.

For all headings 4.1-4.3: You can write as frequently reported rather than commonly used.

After research trends, you need to add an independent heading of future research. I can see some lines in the conclusion, but these are not enough. Move those lines up and make an independent heading. Think of the future research areas and suggest some key areas. This is the main take away from this review.

General comment discussion section: I can see a lot of similar mistakes which I have pointed out before. So, read it again. Try to discuss your results and the future directions. This is missing in the discussion. Also, the starting sentence of all paragraphs in the discussion section should be revised.   

L509: Start with our review also identified. Don’t use vague phrases.  

Conclusion section: Try to be specific in this section. This part is usually optional and added when your discussion section is too long. So, try to be very specific here.

References: Use some reference citation software like EndNote, Mendeley or Zotero to organize your reference as per the MDPI Insects style.

Author Response

(The authors gave the same response as above.)

Round 2

Reviewer 2 Report

Comments and Suggestions for Authors

The authors have incorporated the majority of the changes. However, the references and the formatting of the headings are still not as per the MDPI format.

The revised title is incorrect: Replace it with: Global Trends in Research on Biological Control Agents of Drosophila suzukii: A Systematic Review

or: Global Research Trends on Biological Control Agents of Drosophila suzukii: A Systematic Review

L276: Table heading: It is not BCA Agents; write BCAs or simply BC Agents.

Author Response

(The authors gave the same response as above.)
